# Compressive Sensing via Variational Bayesian Inference under Two Widely Used Priors: Modeling, Comparison and Discussion

**DOI:** 10.3390/e25030511

**Published:** 2023-03-16

**Authors:** Mohammad Shekaramiz, Todd K. Moon

**Affiliations:** 1Machine Learning & Drone Lab, Electrical and Computer Engineering Program, Engineering Department, Utah Valley University, 800 West University Parkway, Orem, UT 84058, USA; 2Electrical and Computer Engineering Department, Utah State University, 4120 Old Main Hill, Logan, UT 84322, USA

**Keywords:** compressive sensing, signal recovery, variational Bayes inference, sparse Bayesian learning, prior modeling, hyperparameters, graphical Bayesian representation

## Abstract

Compressive sensing is a sub-Nyquist sampling technique for efficient signal acquisition and reconstruction of sparse or compressible signals. In order to account for the sparsity of the underlying signal of interest, it is common to use sparsifying priors such as Bernoulli–Gaussian-inverse Gamma (BGiG) and Gaussian-inverse Gamma (GiG) priors on the components of the signal. With the introduction of variational Bayesian inference, the sparse Bayesian learning (SBL) methods for solving the inverse problem of compressive sensing have received significant interest as the SBL methods become more efficient in terms of execution time. In this paper, we consider the sparse signal recovery problem using compressive sensing and the variational Bayesian (VB) inference framework. More specifically, we consider two widely used Bayesian models of BGiG and GiG for modeling the underlying sparse signal for this problem. Although these two models have been widely used for sparse recovery problems under various signal structures, the question of which model can outperform the other for sparse signal recovery under no specific structure has yet to be fully addressed under the VB inference setting. Here, we study these two models specifically under VB inference in detail, provide some motivating examples regarding the issues in signal reconstruction that may occur under each model, perform comparisons and provide suggestions on how to improve the performance of each model.

## 1. Introduction

Compressive sensing (CS) involves efficient signal acquisition and reconstruction techniques in a sub-Nyquist sampling sense. The CS framework can capture the vital information of the underlying signal via a small number of measurements while retaining the ability to reconstruct the signal. CS operates under the assumption that the signal is compressible or sparse, and the number and location of dominating components are unknown in most cases [1,2,3]. Compressibility or sparsity means that the signal has few dominating elements under some proper basis. CS has been used in a variety of applications such as the single-pixel camera, missing pixels and inpainting removal of images, biomedical such as heart rate estimation, internet of things (IoT), geostatistical data analysis, seismic tomography, communications such as blind multi-narrowband signals sampling and recovery, the direction of arrival (DoA) estimation, spectrum sharing of radar and communication signals, wireless networks and many more [4,5,6,7,8,9,10,11,12,13,14,15,16,17,18,19,20,21,22,23,24,25,26,27]. In the linear CS framework, the problem is posed as
(1)$$y=Ax_{s}+e,$$
where $y\;\in \;R^{M}$ contains the measurements, $x_{s}\;\in \;R^{N}$ is the sparse signal of interest, e is the noise representing either the measurement noise or the insignificant coefficients of $x_{s}$ and, generally, M≪N [1,2]. The measurement matrix can be defined as A=ΦΨ, where Φ is the sensing design matrix and Ψ is a proper sparsifying basis. There exist various approaches to solve for $x_{s}$ in (Equation 1) including greedy-based, convex-based, thresholding-based and sparse Bayesian learning (SBL) algorithms [27,28,29,30,31,32,33,34,35,36,37,38,39,40,41,42,43,44,45,46,47,48,49,50,51,52,53,54,55,56,57,58,59,60,61,62,63,64]. Typically, the performance of CS reconstruction is determined in terms of the mean-squared reconstruction error. In this paper, we are also interested in the more demanding requirements of the probability of detection and the false alarm of the nonzero components. This is of more interest to CS applications such as blind multinarrowband signals, spectrum sharing RADAR, etc. [11,12,13,14,15].

The focus of this paper is on sparse Bayesian learning (SBL) for the CS problem. Bayesian learning models are flexible in incorporating prior knowledge of the characteristics of the underlying signal into the model. Bayesian learning also provides a distribution of the hidden variables, which is more informative than the point estimate approaches. A prior favoring the sparsity or compressibility in $x_{s}$ can be represented in the SBL framework via Gaussian-inverse Gamma (GiG), Laplace-inverse Gamma (LiG), Bernoulli–Gaussian-inverse Gamma (BGiG), often referred to as spike-and-slab prior, etc. [27,46,47,48,49,50,51,52,53,54,55,56,57,58,59]. The inference on parameters and hidden variables in these models is usually made using Markov chain Monte Carlo (MCMC) and variational Bayes (VB) [27,45,46,47,48,49,50,51,52]. In this paper, we focus on the two most commonly used SBL prior models for solving the inverse problem of compressive sensing: Bernoulli–Gaussian-inverse Gamma (BGiG) prior and Gaussian-inverse Gamma (GiG). These models have been widely used, along with some additional priors, for sparse recovery of signals or images with block-sparsity/clustering patterns, sparse signals with correlated coefficients or other structured patterns [26,27,48,49,50,51,62,63].

We use VB inference to estimate the variables and parameters of the SBL model. VB is preferred over MCMC because MCMC is computationally expensive, though it can numerically approximate exact posterior distributions with a sufficient amount of computation. The convergence diagnostic of MCMC requires additional work, such as measuring the potential scale reduction factor (PSRF) for all the hidden variables and parameters of the model or monitoring their trace plots [45,50,51,52,65]. In contrast, VB inference can lead to a reasonable approximation of the exact posteriors, using less computation than MCMC and less effort to monitor the convergence [45,51,66,67]. In this paper, we present the derivation of the update rules of the parameters and variables using VB inference for both the BGiG and GiG models. (Portions of this derivation have been previously presented in [68,69]). Although these prior models have been widely used in various applications of compressive sensing, the study of the overall performance of these models under VB inference has yet to be thoroughly investigated. The preference for one model over the other becomes crucial when dealing with moderate or low sampling ratios, which we discuss in this paper. Here, we study the issues associated with each model via some motivational examples. Pre-/postprocessing approaches will then be discussed to tackle the issues. Finally, the overall performance of BGiG and GiG is compared.

The remainder of this work is organized as follows. In Section 2, we present a brief background on VB inference. We study Bernoulli–Gaussian-inverse Gamma modeling for CS using VB in Section 3. Some motivational examples are provided to show the issues with this approach. Section 4 represents Gaussian-inverse Gamma modeling, the associated update rules using VB inference and a motivational example of the issue that may occur using this approach. In Section 5, we study the improvement of the performances of the models after some pre-/postprocessing along with simulation results and comparisons. Section 6 concludes this work.

## 2. Variational Bayesian Inference

Variational Bayes (VB) is an effective approach to approximate intractable integrals that may arise in Bayesian inference. The main idea behind variational methods is to use a family of distributions over the latent variables with their own variational parameters. VB is a fast alternative to sampling methods such as Markov chain Monte Carlo (MCMC) and Sequential Monte Carlo (SMC) for performing approximate Bayesian inference [70,71]. For a probabilistic model with unknown parameters θ and hidden variables x, the posterior distribution of the unknowns, given a set of observations y, can be written as p(x,θ|y)=p(x,θ,y)/p(y). Finding the exact posterior in closed form to perform the inference would be a challenge, as the marginal distribution p(y)=∫p(y,x,θ)dxdθ is often intractable. As an efficient approximation method for such inference problems, VB provides an analytical approximation to the posterior p(x,θ|y). VB approximates the joint density p(x,θ|y) via a variational distribution $Q_{x,\theta }\left(x,\theta \right)$, i.e., $p\left(x,\theta |y\right)\;\approx \;Q_{x,\theta }\left(x,\theta \right)$. VB assumes that the distribution *Q* can be fully factorized with respect to the unknown parameters and hidden variables, i.e.,
$$\begin{array}{cc} Q_{x,\theta }\left(x,\theta \right) & =q_{x}\left(x\right)q_{\theta }\left(\theta \right)\\ & =\prod_{i=1}^{I}q_{x}\left(x_{i}\right)\prod_{j=1}^{J}q_{\theta }\left(\theta_{j}\right), \end{array}$$
where *I* and *J* are the number of unknown parameters and hidden variables, respectively. This independence assumption in VB further simplifies the search for a closed-form solution to the approximation of the actual posterior. We desire to select the variational distribution $Q_{x,\theta }^{*}\left(x,\theta \right)$ as close as possible to p(x,θ|y), where the closeness metric for distribution $Q_{x,\theta }\left(x,\theta \right)$ is formulated as minimizing the Kullback–Leibler (KL) divergence of the approximation $Q_{x,\theta }\left(x,\theta \right)$ and the true posterior p(x,θ|y) as
$$\begin{array}{cc} Q_{x,\theta }^{🟉}\left(x,\theta \right) & =\underset{Q_{x,\theta }\left(x,\theta \right)}{\arg\;\min}\;\mathrm{KL}\left(Q_{x,\theta }\left(x,\theta \right)\left|\right|p\left(x,\theta |y\right)\right)\\ & =\underset{Q_{x,\theta }\left(x,\theta \right)}{\arg\;\min}\int Q_{x,\theta }\left(x,\theta \right)\log\frac{Q_{x,\theta }\left(x,\theta \right)}{p(x,\theta |y)}dxd\theta . \end{array}$$
The quantity logp(y) can be written as logp(y)=log{∫p(x,θ,y)dxdθ}. Then, defining
$$F\left(Q_{x,\theta }\left(x,\theta \right)\right)=\int Q_{x,\theta }\left(x,\theta \right)\log\frac{p(x,\theta ,y)}{Q_{x,\theta }\left(x,\theta \right)}dxd\theta .$$
It is straightforward to show that
$$\log p(y)=F\left(Q_{x,\theta }\left(x,\theta \right)\right)+\mathrm{KL}\left(Q_{x,\theta }\left(x,\theta \right),p\left(x,\theta |y\right)\right).$$
Since (by Jensen’s inequality) $KL(Q_{x,\theta }\left(x,\theta \right)\geq 0$, $\log(p\left(y\right)\geq F(Q_{x,\theta }\left(x,\theta \right)$. Since log(p(y) is constant with respect to $Q_{x,\theta }$, minimizing the KL-divergence between the actual posterior distribution and the variational distribution is equivalent to maximizing the lower bound F(·) [66,67]. Since the term p(y) in p(x,θ|y)=p(x,θ,y)p(y) does not involve the variational distribution $Q_{x,\theta }\left(x,\theta \right)$, this term can be ignored when maximizing F(·). The lower bound F(·) on the model log-marginal likelihood can be iteratively optimized until the convergence by the following update rules [66,72].

VB-E step: 
(2)$$q_{x}^{\left[t+1\right]}\left(x\right)\;\propto \;\exp\left\{E_{q_{\theta }^{\left[t\right]}}\left[\log p(x,y|\theta )\right]\right\}$$
VB-M step: 
(3)$$q_{\theta }^{\left[t+1\right]}\left(\theta \right)\;\propto \;p\left(\theta \right)\exp\left\{E_{q_{x}^{\left[t+1\right]}}\left[\log p(x,y|\theta )\right]\right\}$$
This results in an iterative algorithm analogous to the expectation-maximization (EM) approach.

## 3. Bernoulli–Gaussian-Inverse Gamma Modeling and SBL(BGiG) Algorithm

In the inverse problem of CS defined in (Equation 1), the goal is to recover the sparse vector $x_{s}$. In the Bernoulli–Gaussian-inverse Gamma model, the sparse solution is defined as
(4)$$x_{s}\;=\;\left(s\circ x\right),$$
where s is a binary support vector indicating the non-zero locations in the solution, x represents values of the solution and ∘ is Hadamard (element-by-element) product [47]. We refer to the algorithm associated with this Bayesian modeling based on VB inference as SBL(BGiG). SBL using VB inference for the clustered pattern of sparse signals has already been investigated in the recent literature [45,50,51,58]. In this paper, however, we intend to focus on the ordinary SBL using VB inference modeling without promoting any structure on the supports other than sparsity itself. We show that when the sampling ratio is moderate or low (with respect to the sparsity level), the reconstruction performance becomes sensitive to selecting the support-related hyperparameters.

We define a set of priors as follows [47,68,69]. We model the elements of vector s as
(5)$$s_{n}\;∼\;\mathrm{Bernoulli}\left(\gamma_{n}\right),\gamma_{n}\;∼\;\mathrm{Beta}\left(\alpha_{0},\beta_{0}\right),\forall n,$$
where $\alpha_{0}$ and $\beta_{0}$ are the support-related hyperparameters. Setting $\alpha_{0}$ and $\beta_{0}$ to small values and with $\alpha_{0}\;\ll \;\beta_{0}$ encourages s to be sparse on average. The prior on the solution value vector is defined as
(6)$$x\;∼\;N\left(0,\tau^{-1}I_{N}\right),\;\tau \;∼\;\mathrm{Gamma}\left(a_{0},b_{0}\right).$$
Here, τ is the precision value. Finally, the prior on the noise is
(7)$$e∼N\left(0,\epsilon^{-1}I_{M}\right),\;\epsilon ∼\mathrm{Gamma}\left(\theta_{0},\theta_{1}\right),$$
where $\theta_{0}$ and $\theta_{1}$ are set to small positive values.

### 3.1. Update Rules of SBL(BGiG) Using VB Inference

According to the VB algorithm defined in (Equation 2) and (Equation 3), the update rule of the variables and parameters of the BGiG model can be simplified as follows [68]. The details of these derivations appear in Section A.1.
Update rule for the support vector s
$$q\left(s_{n}|-\right)∼\mathrm{Bernoulli}\left(\frac{1}{1+c_{n}\kappa_{n}}\right),\;\forall n=1,\ldots ,N,$$
where conditioning on − denotes conditioning on all relevant variables and observations. Therefore,
(8)$${\tilde{s}}_{n}\;=\;\frac{1}{1\;+\;c_{n}\kappa_{n}},\;\forall n=1,\ldots ,N,$$
where
(9)$$\begin{array}{cc} c_{n} & :=e^{\psi \left(\beta_{1,n}\right)-\psi \left(\alpha_{1,n}\right)},\\ \kappa_{n} & \;:=\;e^{\frac{1}{2}\tilde{\epsilon }\left(∥a_{n}{∥}_{2}^{2}\left({\tilde{x_{n}}}^{2}\;+\;\sigma_{\tilde{x_{n}}}^{2}\right)\;-\;2\tilde{x_{n}}a_{n}^{T}{\tilde{y}}^{-n}\right)},\\ {\tilde{y}}_{m}^{-n} & \;:=\;y_{m}\;-\;\sum_{l\neq n}^{N}a_{ml}{\tilde{s}}_{l}{\tilde{x}}_{l}. \end{array}$$
Here, $\tilde{x}:=<x{>}_{q_{x}}$, ψ is the digamma function (the logarithmic derivative of the gamma function), and ${\tilde{y}}^{-n}\;=\;{\left[{\tilde{y}}_{1}^{-n},\ldots ,{\tilde{y}}_{M}^{-n}\right]}^{T}$.
Update rule for the solution value matrix x
$$q\left(x|-\right)∼N\left(\tilde{x},\Sigma_{\tilde{x}}\right),$$
where
(10)$$\Sigma_{\tilde{x}}={\left(\tilde{\tau }I_{N}+\tilde{\epsilon }\tilde{\Phi }\right)}^{-1}\;\mathrm{and}\;\tilde{x}=\tilde{\epsilon }\Sigma_{\tilde{x}}\mathrm{diag}\left(\tilde{s}\right)A^{T}y,$$
and where diag(s) denotes a diagonal matrix with the components of s on its main diagonal, and
(11)$$\tilde{\Phi }\;:=\;\left[\left(A^{T}A\right)\circ \left(\tilde{s}{\tilde{s}}^{T}\;+\;\mathrm{diag}\left(\tilde{s}\circ \left(1-\tilde{s}\right)\right)\right)\right].$$
Update rule for $\gamma_{n}$
$$q\left(\gamma_{n}|-\right)∼\mathrm{Beta}\left(\alpha_{1,n},\beta_{1,n}\right),\;\forall n=1,\ldots ,N.$$
Therefore,
(12)$$\tilde{\gamma_{n}}=\frac{\alpha_{1,n}}{\alpha_{1,n}+\beta_{1,n}},\;\forall n=1,\ldots ,N,$$
where
(13)$$\alpha_{1,n}:=\alpha_{0}+{\tilde{s}}_{n}\;\mathrm{and}\;\beta_{1,n}:=\beta_{0}+1-{\tilde{s}}_{n}.$$

Update rule for the solution precision τ$$q\left(\tau |-\right)∼\mathrm{Gamma}\left(a_{0}+\frac{N}{2},b_{0}+\frac{1}{2}(∥\tilde{x}{∥}_{2}^{2}+\mathrm{Tr}\left(\Sigma_{\tilde{x}}\right))\right),$$
where $\Sigma_{\tilde{x}}\;=\;\mathrm{diag}\left(\sigma_{{\tilde{x}}_{1}}^{2},\ldots ,\sigma_{{\tilde{x}}_{N}}^{2}\right)$ and Tr(A) is the trace of matrix *A*. Thus
(14)$$\tilde{\tau }=\frac{a_{0}+\frac{N}{2}}{b_{0}+\frac{1}{2}\left(∥\tilde{x}{∥}_{2}^{2}+\sum_{n=1}^{N}\sigma_{{\tilde{x}}_{n}}^{2}\right)}.$$
Update rule for the noise precision ϵ
$$q\left(\epsilon |-\right)∼\mathrm{Gamma}\left(\theta_{0}+\frac{M}{2},\theta_{1}+\frac{1}{2}\tilde{\Psi }\right),$$
where
(15)$$\tilde{\Psi }\;:=\;\left(y^{T}y-2{\left(\tilde{x}\circ \tilde{s}\right)}^{T}A^{T}y+\mathrm{Tr}\left(\left(\tilde{x}{\tilde{x}}^{T}+\Sigma_{\tilde{x}}\right)\tilde{\Phi }\right)\right).$$
This yields to the following update rule for the precision of the noise component
(16)$$\tilde{\epsilon }=\frac{\theta_{0}+\frac{M}{2}}{\theta_{1}+\frac{1}{2}\tilde{\Psi }}.$$

The stopping criterion of the algorithm is made based on the log-marginalized likelihood. We define the stopping condition in terms of L:=log{p(y|s,ϵ,τ)}. The marginalized likelihood can be written as
$$p\left(y|s,\epsilon ,\tau \right)\;=\;\int p\left(y|x,s,\epsilon \right)p\left(x|\tau I_{N}\right)dx.$$
After some simplification, the negative log-likelihood is proportional to
$$-L\;\propto \;\log|\Sigma_{0}^{-1}|+y^{T}\Sigma_{0}y,$$
where
(17)$$\Sigma_{0}\;=\;{\left({\tilde{\epsilon }}^{-1}I_{M}\;+\;{\tilde{\tau }}^{-1}A{\tilde{S}}^{2}A^{T}\right)}^{-1}$$
and $\tilde{S}\;:=\;\mathrm{diag}\left\{\tilde{s}\right\}$. Therefore, the stopping condition can be made as
(18)$$\Delta L_{n}^{\left[t\right]}:=|\Delta L^{\left[t\right]}\left|/\right|L^{\left[t-1\right]}|\leq T_{0},$$
for some small value of threshold $T_{0}$ [50], where
(19)$$L^{\left[t\right]}:=\log\Sigma_{0}^{\left[t\right]}-y^{T}\Sigma_{0}^{\left[t\right]}y.$$
and
(20)$$\begin{array}{cc} \Delta L^{\left[t\right]}: & =L^{\left[t\right]}-L^{\left[t-1\right]}\\ \; & =\log|\frac{\Sigma_{0}^{\left[t\right]}}{\Sigma_{0}^{\left[t-1\right]}}|+y^{T}\left(\Sigma_{0}^{\left[t-1\right]}-\Sigma_{0}^{\left[t\right]}\right)y. \end{array}$$

Figure 1 illustrates the graphical Bayesian representation of the BGiG model, which is an undirected graph. The shaded node y shows the observations (measurements), and the small solid nodes represent the hyperparameters. Each unshaded node denotes a random variable (or a group of random variables).

The flowchart representation of the algorithm is shown in Figure 2 motivated by the graphical approach in [47,73]. According to the pseudocode in Algorithm 1 and the flowchart in Figure 2, first, the hyperparameters of the model are set. The support-related hyperparameters $\alpha_{0}$ and $\beta_{0}$ are suggested to be set to small values with $\alpha_{0}\;\ll \;\beta_{0}$ to encourage s to be sparse on the average. The hyperparameters $a_{0}$ and $b_{0}$ on the precision of the solution-value vector are also initialized and suggested to be small not to bias the estimation when the measurements are incorporated. The hyperparameters $\theta_{0}$ and $\theta_{1}$ on the precision of the noise are recommended to be of order $10^{-6}$ for high SNRs. For moderate and low SNRs, higher values are recommended. In the next step, all the main variables of the model are drawn i.i.d. from their corresponding prior distributions defined in (Equation 5)–(Equation 7). Then, the stopping condition is computed based on the log-marginalized likelihood in (Equation 19). In the main loop, all of the main variables of the model are updated via the expected values obtained from the VB inference. Specifically, we first update the support vector and the solution value components; then, the precisions of the solution vector and the noise are updated. Finally, the stopping criterion is computed through the measure of the log-marginalized likelihood of the observations. The pseudocode of the algorithm is provided below.
**Algorithm 1:** SBL(BGiG) Algorithm${\hat{x}}_{s}=\tilde{x}\circ \tilde{s}$$\left[\tilde{x},\tilde{s}\right]=\mathrm{SBL}-\mathrm{BGiG}\left(Y,A\right)$Set the hyperparameters, i.e., ($\alpha_{0},\beta_{0}$), ($a_{0},b_{0}$), and ($\theta_{0},\theta_{1}$)% Variables InitializationDraw $\tilde{s}$ and $\tilde{\gamma }$ from (Equation 5)Draw $\tilde{x}$ and $\tilde{\tau }$ from (Equation 6)Draw $\tilde{\epsilon }$ from (Equation 7)t = 1     % IteratorCompute $L^{\left[t\right]}$ from (Equation 19) and set $L^{\left[0\right]}=0$% Main Loop for Estimations**While**$\frac{|L^{\left[t\right]}-L^{\left[t-1\right]}|}{|L^{\left[t-1\right]}|}\geq T_{0}$. For example $T_{0}=10^{-6}$.       Compute ${\tilde{s}}_{n}$ from (Equation 8), ∀n=1,…,N                         % (Support vector component )       Compute $\Sigma_{\tilde{x}}$ and $\tilde{x}$ from (Equation 10)                                   % (Solution-value matrix component)       Compute $\alpha_{1,n}$ and $\beta_{1,n}$ from (Equation 13) ∀n=1,…,N        % (Parameters of the hyperprior γ)       t Compute $\tilde{\tau }$ from (Equation 14)                                         % (Precision on the solution)       Compute $\tilde{\epsilon }$ from (Equation 16)                                            % (Precision on the noise)       Compute $L^{\left[t\right]}$ from (Equation 19) and then t = t + 1**End While**

### 3.2. Issues with SBL(BGiG)

In this section, we show that the estimated solution using SBL(BGiG) algorithm is sensitive to support-related hyperparameters, i.e., $\alpha_{0}$ and $\beta_{0}$ in (Equation 5). We provide an example under three cases to demonstrate this issue. We generated a random scenario, where the true solution $x_{s}\;\in \;R^{100}$ has the sparsity level of k=25, that is, the true x (or s) has *k* active elements. The active elements of s were drawn randomly. The nonzeros of $x_{s}$, corresponding to the active locations of s, were drawn from $N(0,\sigma_{x}^{2})$, with $\sigma_{x}^{2}=1$. Each entry of the sensing matrix *A* was drawn i.i.d. from the Gaussian distribution N(0,1), then normalized, so each column has the Euclidian norm of 1. The elements of measurement noise were drawn from $N(0,\sigma^{2})$ with SNR=25 dB, where $SNR:=\;20\log_{10}\left(\sigma_{x}/\sigma \right)$. The hyperparameters of τ and ϵ were set to $a_{0}\;=\;b_{0}\;=\;10^{-3}$ and $\theta_{0}\;=\;\theta_{1}\;=\;10^{-6}$, respectively. In Cases 1–3, we set the pair $\left(\alpha_{0},\beta_{0}\right)$ with low emphasis on the prior (0.01,0.99), moderate emphasis (0.1,0.9) and fairly high emphasis (1.4,2), respectively.

From the top to the bottom row of Figure 3, Figure 4 and Figure 5, we illustrate the estimated results with the number of measurements set to 80, 60 and 40 (that is, the sample ratio λ is 0.80, 0.60, and 0.40), respectively. In each row of Figure 3, Figure 4 and Figure 5 from left to right, we show the comparison between the measurements y and the computed measurements based on $\hat{y}\;=\;A\left(\tilde{s}\;\circ \;\tilde{x}\right)$, the true signal $x_{s}\;=\;s\circ x$ and the reconstructed signal ${\hat{x}}_{s}\;=\;\tilde{s}\circ \tilde{x}$, the true support vector s and the estimated support vector $\tilde{s}$ and the evolution of the estimated supports with respect to the iterations in the SBL(BGiG) algorithm.

According to Figure 3, the setting for $\left(\alpha_{0},\beta_{0}\right)$ in Case 1 fails to provide perfect results even for high sampling ratios. Similarly, Figure 4 shows that the settings for $\left(\alpha_{0},\beta_{0}\right)$ in Case 2 do not provide encouraging results even for high sampling ratios. Specifically, it turns out that Case 1 and Case 2 provide sparse solutions for the sampling ratios within the range [0,1], where λ=1 means M=N.

According to Figure 5, setting $\left(\alpha_{0},\beta_{0}\right)$ to (1.4,2) seems to be a reasonable choice for high sampling ratios (over 70%), while it is not a good choice for the lower sampling ratios. This issue can be seen in the supports plot in the 2nd and 3rd row of Figure 5. One may argue that the estimated support vector $\hat{s}$ can be filtered via some threshold value (such as 0.3) for λ=0.6. However, thresholding will adversely affect the detection rate, and setting the threshold depends on our understanding of the signal characteristics. Furthermore, we should account for the effect of the filtered supports since their corresponding estimated components in ${\hat{x}}_{s}$ contribute to fitting the model to the measurements.

In Table 1, we summarize the performance of the generated example for Cases 1–3, where $P_{D}$, $P_{FA}$ and NMSE denote the detection rate and false alarm rate in support recovery and the normalized mean-squared error between the true and the estimated sparse signal. This also shows that the algorithm fails to provide reasonable results for the sampling ratio of λ=0.4.

These experiments suggest that there is no fixed setting for $\left(\alpha_{0},\beta_{0}\right)$ capable of performing reasonably well for all sampling ratios and thus, selecting the hyperparameters $\left(\alpha_{0},\beta_{0}\right)$ should be made with care.

Continuing this examination, in Figure 6, Figure 7 and Figure 8, we illustrate the negative log-marginalized likelihood, the noise precision estimation and the estimated precision on the generated true solution in Cases 1–3, respectively. The horizontal axis shows the iterations until the stopping rule is met.

As expected, as the sampling ratio increases, the algorithm requires fewer iterations to meet its stopping condition. This can be seen on the negative log-marginalized likelihood plots in Figure 6, Figure 7 and Figure 8. In these experiments, the actual precision of the solution components was set to τ=1, and the actual noise precision was set to ϵ=316.2.

For Cases 1 and 2, according to Figure 6, Figure 7 and Figure 8, the estimated precisions on both the noise and solution components were far off from the actual ones even for λ=0.8. Thus, it resulted in poor performance in signal recovery for Cases 1 and 2 (see Figure 3 and Figure 4).

For Case 3, the estimated precisions on the noise and the solution components were acceptable for λ=0.8 but far off from the actual ones for lower sampling ratios (see Figure 8). The main issue of the failures can be found in the update rule of the support learning vector $\tilde{s}$ defined in (Equation 8). It is important to balance between the terms $c_{n}$ and $\kappa_{n}$, where $c_{n}$ imposes the effect of hyper-prior on s accompanied by the current estimate of $s_{n}$. In contrast, $\kappa_{n}$ imposes the contribution of the current estimates of noise precision, solution and other supports in fitting the model to the measurements. Therefore, if we impose a substantial weight on the sparsity via $c_{n}$, the solution tends to neglect the effect of $\kappa_{n}$ and vice versa. This is why we had sparse (with poor performance) in Cases 1 and 2 for all the represented sampling ratios and nonsparse (with poor performance) for moderate and lower sampling ratios in Case 3. These results suggest that the algorithm and its update rules are sensitive to the selection of hyperparameters on the Gamma prior on the support vector s. The main issue can be seen in (Equation 9), where the selection of the hyperparameters $\alpha_{0}$ and $\beta_{0}$ resulted in a large or small value in $c_{n}$ due to the digamma function.

## 4. Gaussian-Inverse Gamma Modeling and SBL(GiG) Algorithm

In this section, we consider the Gaussian-inverse Gamma (GiG) model. In this model, each component $x_{n}$ of the solution is modeled by zero-mean Gaussian with the precision $\tau_{n}$. The main difference between this model and the model defined in Section 3 is that the GiG model does not have the support vector s; instead, different precisions are considered on the components of the solution vector $x_{s}$ in (Equation 1). A simpler version of GiG can also be used by defining the same precision τ for all the components of $x_{s}$.

Here, we rather use different precisions to make the GiG model have almost the same complexity as the BGiG model in terms of the parameters to be learned. The set of priors in this model is defined as follows.
(21)$$x_{n}\;∼\;N\left(0,\tau_{n}^{-1}\right),\;\tau_{n}\;∼\;\mathrm{Gamma}\left(a_{0},b_{0}\right),\;\forall n,$$
where $a_{0}$ and $b_{0}$ denote the shape and rate of the Gamma distribution, respectively. The entries of the noise component e are defined the same as (Equation 7), i.e., 
$$e∼N\left(0,\epsilon^{-1}I_{M}\right),\;\epsilon ∼\mathrm{Gamma}\left(\theta_{0},\theta_{1}\right),$$
where $\theta_{0}$ and $\theta_{1}$ are set to small positive values. The estimation of the parameters in this model is carried out using VB inference, as discussed below.

### 4.1. Update Rules of SBL(GiG) Using VB Inference

According to the VB algorithm described in (Equation 2) and (Equation 3), the update rule of the variables and parameters of the GiG model can be simplified as follows. The details of these derivations appear in Section A.2.
Update rule for the precision $\tau_{n}$ on $x_{n}$ using VB
$$q\left(\tau_{n}\right)\;∼\;\mathrm{Gamma}\left(a_{0}+\frac{1}{2},b_{0}+\frac{1}{2}\left({\tilde{x}}_{n}^{2}+\sigma_{{\tilde{x}}_{n}}^{2}\right)\right),\;\forall n=1,\ldots ,N.$$
Thus,
(22)$$\tilde{\tau_{n}}=\frac{a_{0}+\frac{1}{2}}{b_{0}+\frac{1}{2}\left({\tilde{x}}_{n}^{2}+\sigma_{{\tilde{x}}_{n}}^{2}\right)},\;\forall n\;=1,2,\ldots ,N.$$
Update rule for the noise precision ϵ using VB
$$q\left(\epsilon \right)\;∼\;\mathrm{Gamma}\left(\theta_{0}+\frac{M}{2},b_{0}+\frac{1}{2}\tilde{\Psi }\right)$$
which yields
(23)$$\tilde{\epsilon }=\frac{\theta_{0}+\frac{M}{2}}{\theta_{1}+\frac{1}{2}\tilde{\Psi }},$$
where
(24)$$\tilde{\Psi }:=y^{T}y-2{\tilde{x}}^{T}A^{T}y+\mathrm{Tr}\left(\left(\tilde{x}{\tilde{x}}^{T}+\Sigma_{\tilde{x}}\right)A^{T}A\right).$$
Update rule for the solution vector x using VB
(25)$$q_{x}\left(x\right)∼N\left(\tilde{x},\Sigma_{\tilde{x}}\right),$$
where
(26)$$\Sigma_{\tilde{x}}:={\left(\tilde{T}+\tilde{\epsilon }A^{T}A\right)}^{-1}\;\mathrm{and}\;\tilde{x}:=\tilde{\epsilon }\Sigma_{\tilde{x}}A^{T}y,$$
and
$$\tilde{T}:=\mathrm{diag}\left\{\left[{\tilde{\tau }}_{1},\ldots ,{\tilde{\tau }}_{N}\right]\right\}.$$

We set the stopping rule of the algorithm using the marginalized likelihood (evidence) defined as
p(y|ϵ,τ)=∫p(y|x,ϵ,τ)p(x|τ)dx.
After simplification and for the comparison purposes of $L^{\left[t\right]}$ with $L^{\left[t-1\right]}$ in the updating process, we have
$$L^{\left[t\right]}\propto \log|\Sigma_{0}^{\left[t\right]}|-y^{T}\Sigma_{0}^{\left[t\right]}y,$$
where $\Sigma_{0}$ is defined as
(27)$$\Sigma_{0}:=\;{\left({\tilde{\epsilon }}^{-1}I_{M}\;+\;{\tilde{T}}^{-1}AA^{T}\right)}^{-1}.$$

Therefore, similar to SBL(BGiG), the stopping condition can be made as
(28)$$\Delta L_{n}^{\left[t\right]}:=|\Delta L^{\left[t\right]}\left|/\right|L^{\left[t-1\right]}|\leq T_{0},$$
for some small value of threshold $T_{0}$.

Figure 9 illustrates the graphical Bayesian representation of the GiG model, which is an undirected graph. Similar to Figure 1, the shaded node y shows the observations, the small solid nodes represent the hyperparameters and the unshaded nodes denote the random variables.

The flowchart representation of the algorithm is shown in Figure 10. According to the pseudocode in Algorithm 2 and the flowchart in Figure 10, first, the hyperparameters of the model are set. The hyperparameters $a_{0}$ and $b_{0}$ on the precision of the solution-value vector are initialized and suggested to be small. Similar to SBL(BGiG), the hyperparameters $\theta_{0}$ and $\theta_{1}$ on the precision of the noise are recommended to be of order $10^{-6}$ for high SNRs. All the main variables of the model are drawn i.i.d. from their corresponding prior distributions defined in (Equation 22)–(Equation 26). Then, the stopping condition is computed based on (Equation 28). In the main loop, all the main variables of the model are updated via the expected values obtained from the VB inference through (Equation 22)–(Equation 26). The pseudocode of the algorithm is provided below.      
**Algorithm 2:** SBL(GiG) Algorithm${\tilde{x}}_{s}=\mathrm{SBL}-\mathrm{GiG}\left(Y,A\right)$                Set the hyperparameters, i.e., ($a_{0},b_{0}$) and ($\theta_{0},\theta_{1}$)% Variables’ InitializationDraw ${\tilde{x}}_{s}$ and $\tilde{ø}$ from (Equation 21)Draw $\tilde{\epsilon }$ from (Equation 7)t = 1     % IteratorCompute ${\tilde{L}}^{\left[t\right]}$ from (Equation 28) and (Equation 27), and set ${\tilde{L}}^{\left[0\right]}=0$% Main Loop for Estimationst = 1**While**$\frac{|L^{\left[t\right]}-L^{\left[t-1\right]}|}{|L^{\left[t-1\right]}|}\geq T_{0}$. For example $T_{0}=10^{-6}$.       Compute $\Sigma_{\tilde{x}}$ and ${\tilde{x}}_{s}$ from (Equation 26)          % (Solution-value matrix component)       Compute $\tilde{T}$ from (Equation 22)                    % (Precisions on the solution)       Compute $\tilde{\epsilon }$ from (Equation 23)                     % (Precision on the noise)       Compute $L^{\left[t\right]}$ from (Equation 28) and (Equation 27), and then t = t + 1**End While**

### 4.2. Issues with SBL(GiG)

An issue with the SBL(GiG) algorithm is that the solution becomes nonsparse since it does not incorporate a binary vector s (hard-thresholding or soft-thresholding if the expected value is used) as we had in SBL(BGiG). This may have no major effect on the signal reconstruction for high sampling ratios. However, the nonsparseness effect appears in low sampling ratios by misleading the algorithm to wrongly activate many components in the estimated signal yet providing a good fit of the model to the measurements. Here, we use the same example as we made for the SBL(BGiG) model with the same sensing matrix *A*, measurement vector y and noise e. Notice that in the SBL(BGiG) model, we considered the same precision τ on all the components of the solution value vector x support vector s. In contrast, the SBL(GiG) model does not have the support learning vector; instead, we assume that each component of the solution vector has different precision $\tau_{n}$. It turns out that SBL(GiG) is not very sensitive to the selection of the hyperparameters as the SBL(BGiG). Thus, here, we show the results for one case scenario for the hyperparameters. We use the same setting for the parameters of ϵ in the hyper prior as before, i.e., $\theta_{0}\;=\;\theta_{1}\;=\;10^{-6}$, and the same parameters for all the precisions $\tau_{n}$ of the solution component, i.e., $a_{0}\;=\;b_{0}\;=\;10^{-3}$. In Figure 11 and Figure 12, we illustrate the results after applying the SBL(GiG) algorithm. In Figure 11, from left to right, we show the results for sampling ratios of λ=0.8, 0.6, and 0.40, respectively. The first row shows the comparison of y with $\hat{y}\;=\;A{\tilde{x}}_{s}$, the second row shows the true solution $x_{s}$ and the estimated solution ${\tilde{x}}_{s}$, and the third row demonstrates the estimated precisions on the solution components. In Figure 12, we demonstrate the negative log-marginalized likelihood comparison and the estimated noise precision against the true noise precision for the sampling ratios of λ= 0.8, 0.6 and 0.4.

From the results shown in Figure 11 and Figure 12, we observe that the recovered signal tends to become nonsparse. This effect is illustrated in the second row of Figure 11. This can also be observed in the precision estimations of the solution components. More specifically, the true nonzero components in our simulations were drawn from a zero-mean Gaussian with the precision of $\tau_{n}\;=\;1$. Thus, the ideal precision estimation would be within the two classes of values of 1 and infinity or very large values. However, the estimated results in our simulation do not show such a classification. As the sampling ratio decreases, the solution estimate has poor performance, due not only to the reduction in the number of measurements but also the nonsparseness behavior.

## 5. Preprocessing versus Postprocessing and Simulations

In this section, we show that in order to improve the performance of Bernoulli–Gaussian-inverse Gamma modeling using the SBL(BGiG) algorithm, we need to perform a preprocessing step. The results in Section 4 suggest one can perform some postprocessing for the SBL(GiG) algorithm to improve the reconstruction performance. Below, we provide more details for each of these algorithms.

### 5.1. Pre-Processing for the SBL(BGiG) Algorithm

Based on the observations made on the performance of SBL(BGiG) in Section 3.2, we showed that the pair of hyperparameters $(\alpha_{0}$,$\beta_{0})$ should be selected with care. In other words, obtaining good performance with this algorithm needs some preprocessing to assess an appropriate setting for the parameters. For a more rigorous study, here, we perform a grid search on the hyperparameters $\left(\alpha_{0},\beta_{0}\right)$ to see whether we can find some common pattern in selecting these parameters for all sampling ratios. The grid search runs the algorithm for different values of $\alpha_{0}$ and $\beta_{0}$ with the search range of [0.1,2] with the resolution of 0.1. For each $\left(\alpha_{0},\beta_{0}\right)$ within this range, we ran 200 random trials and then averaged the results. The settings of these trials are represented in Table 2.

We generated a random scenario, where the true solution $x_{s}\;\in \;R^{100}$ has the sparsity level of k=25. The active elements of s were drawn randomly. The nonzeros of $x_{s}$ were drawn from $N(0,\sigma_{x}^{2})$, with $\sigma_{x}^{2}=1$. Each entry of the sensing matrix *A* was drawn i.i.d. from the Gaussian distribution N(0,1), then normalized. The elements of measurement noise were drawn from $N(0,\sigma^{2})$ with SNR=25 dB. The results were examined to see what values of $\left(\alpha_{0},\beta_{0}\right)$ provided the highest performance in the detection rate vs. and false alarm rate. The simulation was executed for a range of *sampling ratios* in the range [0.05,1] with the step size of 0.05. The results are demonstrated in Figure 13. In this figure, we also provide the results of performing a random Sobol search for $\left(\alpha_{0},\beta_{0}\right)$. A Sobol sequence is a low discrepancy quasirandom sequence. The two right plots in Figure 13 show the results for the best setting of $\left(\alpha_{0},\beta_{0}\right)$.

It should be clear from Figure 13 that there is no fixed setting for these parameters in order to get the best performance for all sampling ratios. The two plots on the right of Figure 13 illustrate the performance based on the best values of these hyperparameters, which provided the best performance, i.e., tuned hyperparameters. We also examined the grid search results for the top 10 highest performances for each sampling ratio, where performance is in terms of $P_{D}\;-P_{FA}$ and the normalized mean-squared error (NMSE). In Figure 14a, we demonstrate the top 10 highest performances based on NMSE and $P_{D}\;-P_{FA}$ for different sampling ratios. In Figure 14b,c, we illustrate the values of $\left(\alpha_{0},\beta_{0}\right)$, which led to the performances shown in Figure 14a for different sampling ratios. Figure 15 details the top 10 values of $\left(\alpha_{0},\beta_{0}\right)$ vs. sampling ratio.

According to Figure 14b,c, there is no specific pattern for these hyperparameters. Figure 15 also shows that hyperparameters need to be carefully selected.

### 5.2. Post-Processing for the SBL(GiG) Algorithm

Since the SBL(GiG) algorithm does not include the binary support vector s, as SBL(BGiG) possesses, the resulting solution tends to become nonsparse. This leads to a high detection rate for the location of active supports and a high false alarm rate. Thus, as the sampling ratio decreases, there is a high chance that this algorithm overwhelms the locations of the true solution. Therefore, SBL(GiG) requires some postprocessing to discard the components with low amplitudes. This problem becomes of great importance for applications where detecting the correct nonzero components is more crucial than the magnitudes of the nonzeros in the signal. This effect can be seen in Figure 16b. The curves with solid lines in this plot show the detection and false alarm rate in support recovery and the difference between the rates. This issue can be resolved by some postprocessing such as data-driven threshold tuning. That way, the amplitudes in the reconstructed signal with lower values than the threshold can be discarded. For this purpose, we set up 200 random trials, the same way as the one explained for SBL(BGiG), and then evaluate the performance in terms of NMSE by varying the threshold. Figure 16b shows the averaged results of 200 trials. The settings of these trials are represented in Table 3.

In Figure 16a, we observe that the postprocessing does not benefit us so much in terms of the reconstruction error for low and moderate sampling ratios. However, there is a threshold of around 0.25, for which the postprocessing step reduced the reconstruction error by approximately 3 dB. We set the threshold to 0.25 and ran 200 random trials by applying SBL(GiG) and evaluating the performance based on the detection and false alarm rate in support recovery. According to Figure 16b, the additional post-processing step provides reasonable performance.

Finally, in Figure 17, we compare the performance of the SBL(BGiG) algorithm (with performing the preprocessing step) with the SBL(GiG) algorithm (after performing postprocessing). We see that Bernoulli–Gaussian-inverse Gamma implemented via SBL(BGiG) provides better performance for low and high sampling ratios. In contrast, Gaussian-inverse Gamma modeling implemented via SBL(GiG) performs much better for the moderate sampling ratios.

## 6. Conclusions

We investigated solving the inverse problem of compressive sensing using VB inference for two sparse Bayesian models of Bernoulli–Gaussian-inverse Gamma (BGiG) and Gaussian-inverse Gamma (GiG). The issues of each approach were discussed and the performance between the two models was compared. Specifically, we showed the behavior of these models and algorithms when the sampling ratio is low and moderate as well as the importance of selecting the hyperparameters of BGiG model with care. We further provided some intuition for performing additional pre/post-processing steps, depending on the selected model for better performance.

Based on our study on the synthetic data and considering the overall performance of both algorithms and the complexity in additional pre-/postprocessing, we observed that for moderate sampling ratios, SBL(GiG) is performing better than SBL(BGiG) modeling when using VB for sparse signals with no specific pattern in the supports. In contrast, SBL(BGiG) provided better perfomance for low and high sampling ratios. Finally, a rigorous comparison is required to study in the future under real-world scenarios and various applications. The MATLAB codes for GiG and BGiG modeling are available at https://github.com/MoShekaramiz/Compressive-Sensing-GiG-versus-BGiG-Modeling.git, accessed on 15 December 2022.

## Figures and Tables

**Figure 1 entropy-25-00511-f001:**
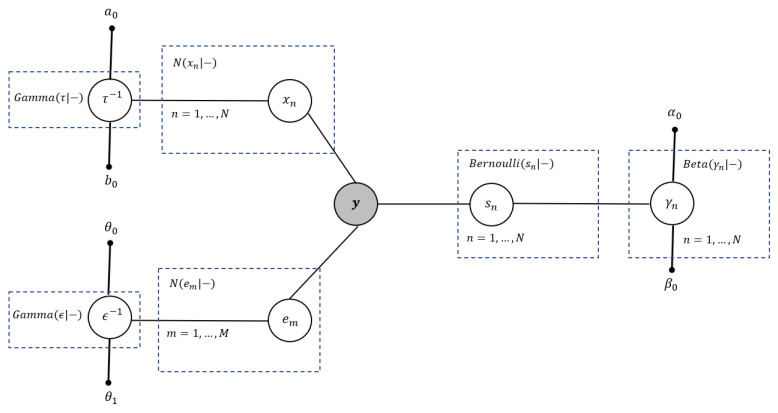
Graphical Bayesian representation of the BGiG model.

**Figure 2 entropy-25-00511-f002:**
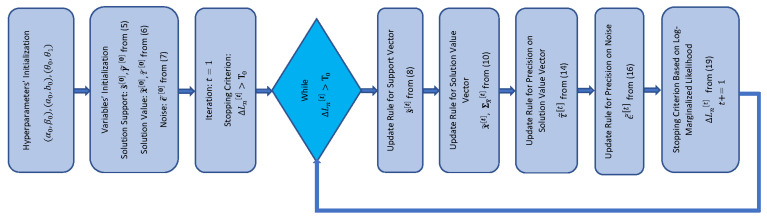
Flowchart of SBL(BGiG) algorithm.

**Figure 3 entropy-25-00511-f003:**
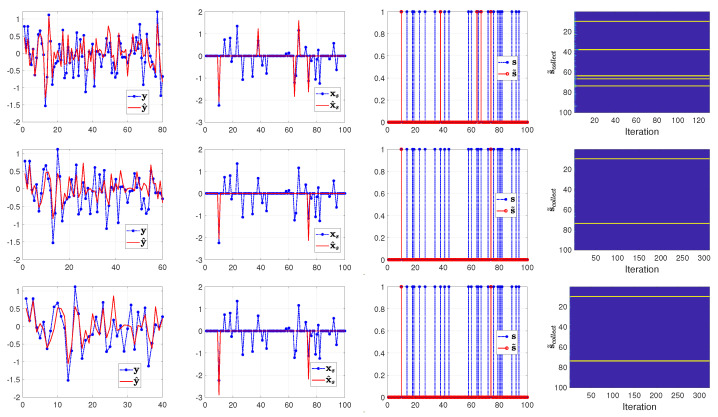
Case 1: $\left(\alpha_{0},\beta_{0}\right)=\left(0.01,0.99\right)$. From top to bottom, the rows show the results of SBL(BGiG) for the sampling ratio λ=0.80,0.60,0.40, respectively.

**Figure 4 entropy-25-00511-f004:**
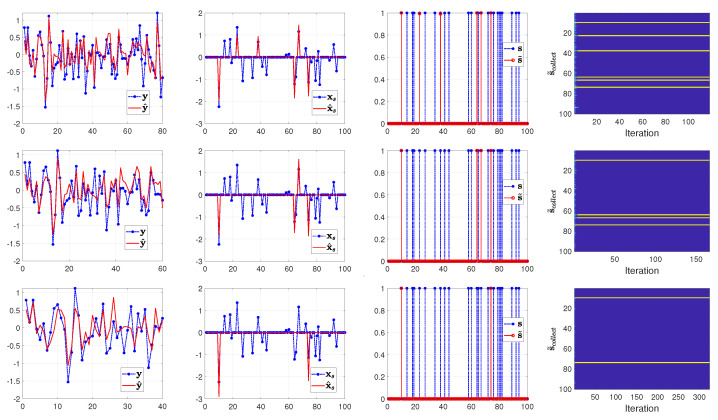
Case 2: $\left(\alpha_{0},\beta_{0}\right)=\left(0.1,0.9\right)$. From top to bottom, the rows show the results of SBL(BGiG) for the sampling ratio λ=0.80,0.60,0.40, respectively.

**Figure 5 entropy-25-00511-f005:**
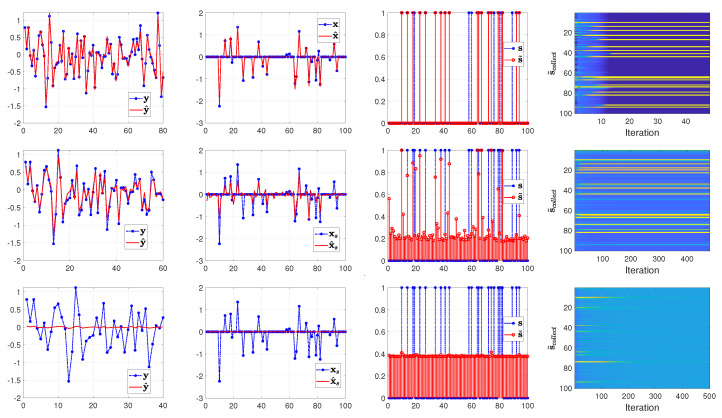
Case 3: $\left(\alpha_{0},\beta_{0}\right)=\left(1.4,2\right)$. From top to bottom, the rows show the results of SBL(BGiG) for the sampling ratio λ=0.80,0.60,0.40, respectively.

**Figure 6 entropy-25-00511-f006:**
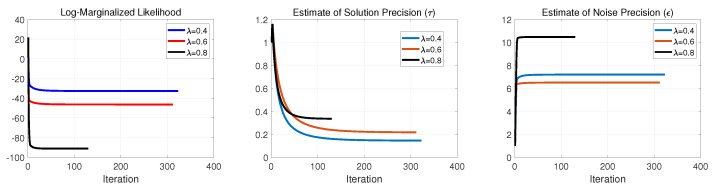
Case 1: Performance evaluation of SBL(BGiG).

**Figure 7 entropy-25-00511-f007:**
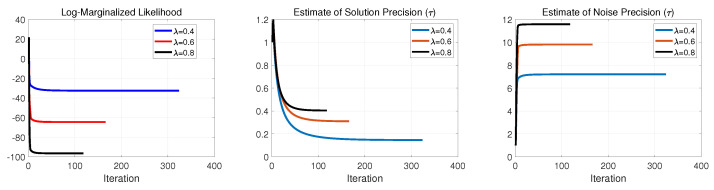
Case 2: Performance evaluation of SBL(BGiG).

**Figure 8 entropy-25-00511-f008:**
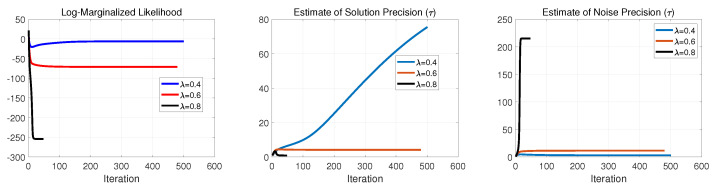
Case 3: Performance evaluation of SBL(BGiG).

**Figure 9 entropy-25-00511-f009:**
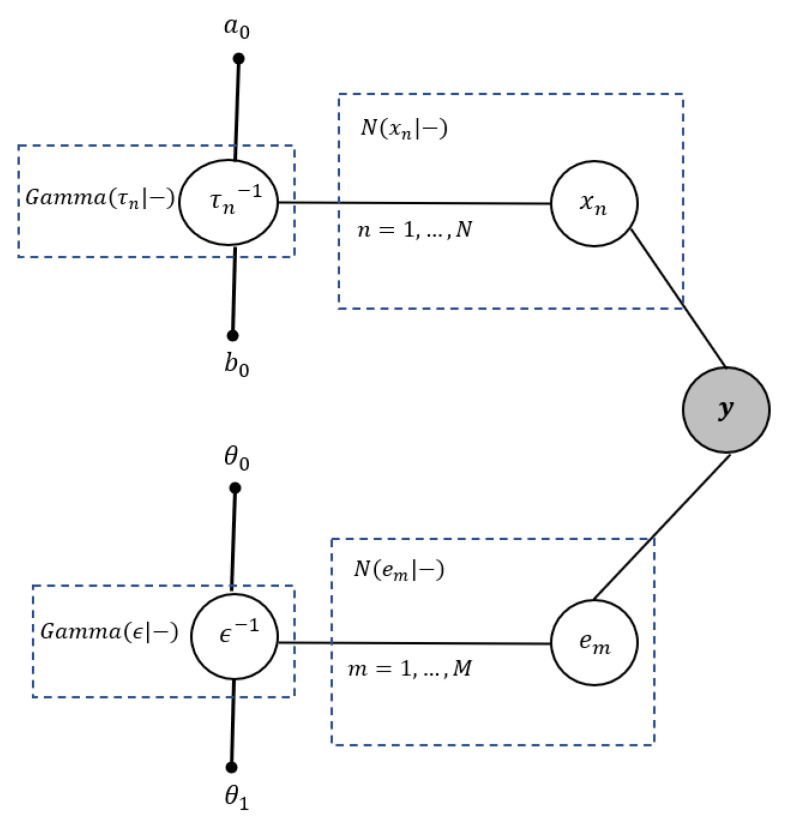
Graphical Bayesian representation of the GiG model.

**Figure 10 entropy-25-00511-f010:**
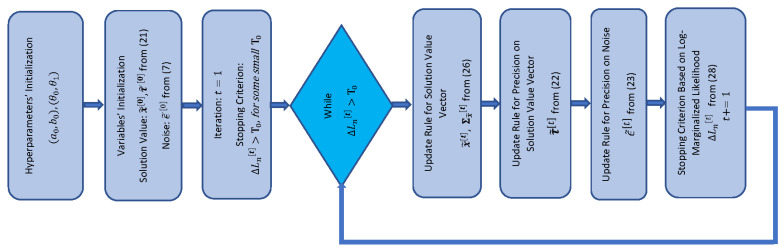
Flowchart of SBL(GiG) algorithm.

**Figure 11 entropy-25-00511-f011:**
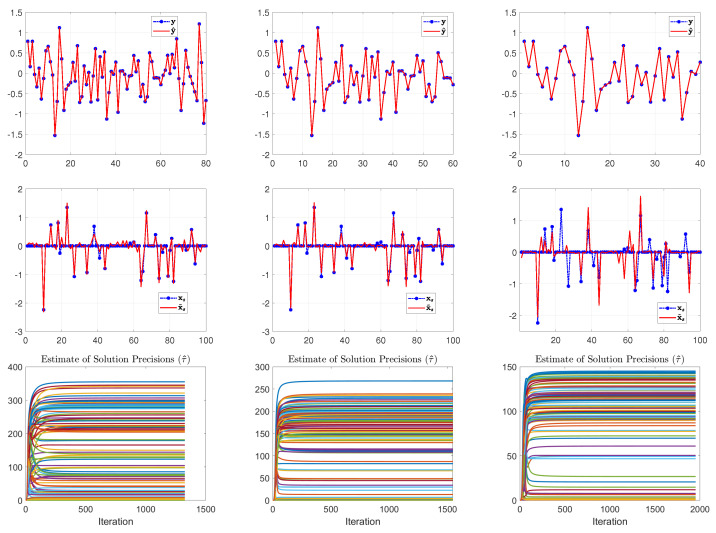
From left to right, we show the results for sampling ratios of λ= 0.8, 0.6 and 0.40, respectively. The first row shows the comparison of y with $\hat{y}\;=\;A\tilde{x_{s}}$, the second row shows the true solution $x_{s}$ and the estimated solution ${\tilde{x}}_{s}$, and the third row demonstrates the estimated precisions on the solution components.

**Figure 12 entropy-25-00511-f012:**
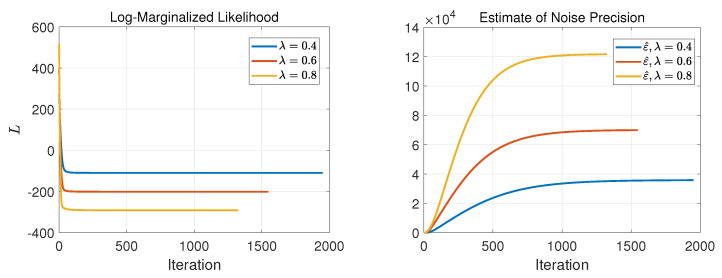
The behavior of negative marginalized log-likelihood and the precision on the noise using SBL(GiG) for the sampling ratios of 0.4, 0.6 and 0.80.

**Figure 13 entropy-25-00511-f013:**
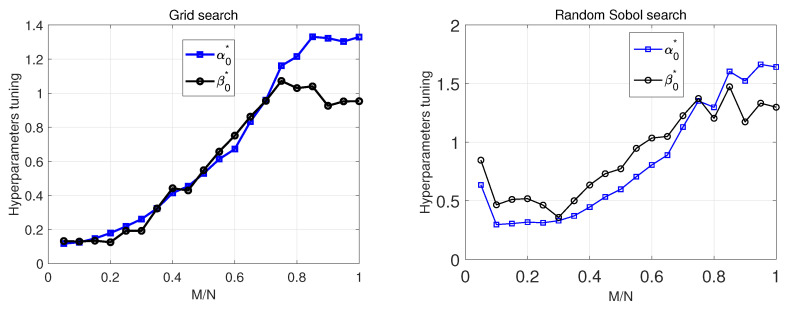
Performance evaluation of SBL(BGiG) using grid and random Sobol search.

**Figure 14 entropy-25-00511-f014:**
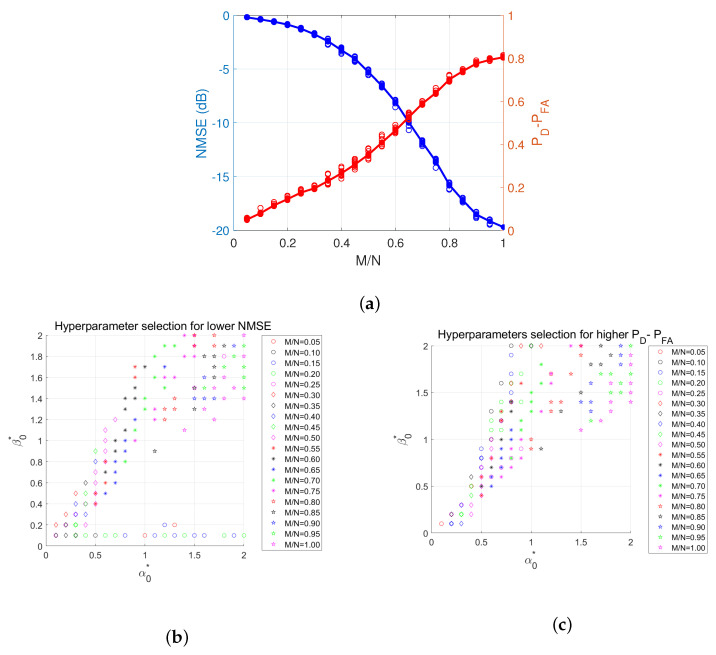
(**a**) Overall performance (**b**) Top 10 $\left(\alpha_{0},\beta_{0}\right)$ with lowest NMSE (**c**) Top 10 $\left(\alpha_{0},\beta_{0}\right)$ with highest $P_{D}\;-\;P_{FA}$.

**Figure 15 entropy-25-00511-f015:**
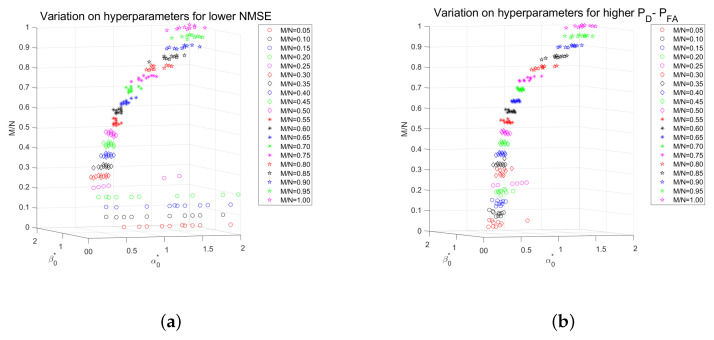
(**a**) Top 10 $\left(\alpha_{0},\beta_{0}\right)$ with lowest NMSE vs. sampling ratio (**b**) Top 10 $\left(\alpha_{0},\beta_{0}\right)$ with highest $P_{D}\;-\;P_{FA}$ vs. sampling ratio.

**Figure 16 entropy-25-00511-f016:**
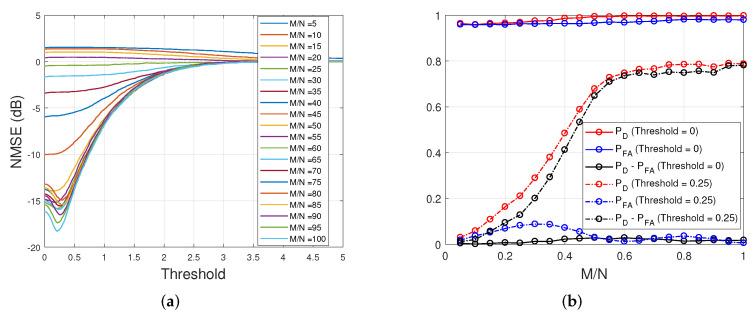
Performance of SBL(GiG). (**a**) NMSE of SBL(GiG) vs. threshold. (**b**) Performance of SBL(GiG) before and after postprocessing.

**Figure 17 entropy-25-00511-f017:**
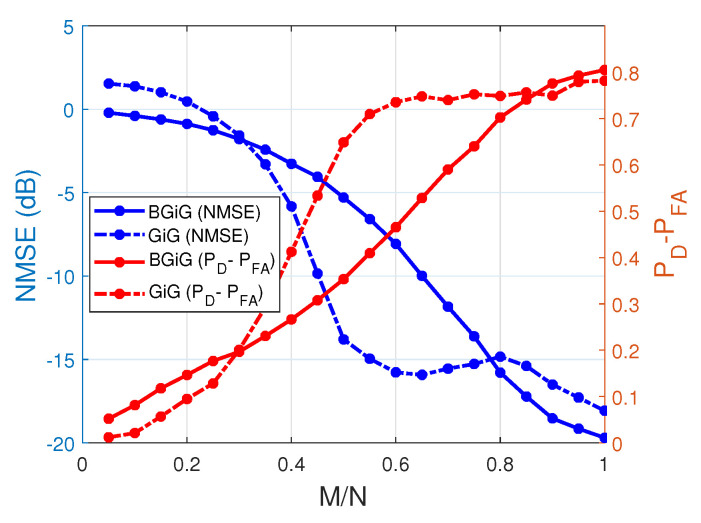
Performance of SBL(BGiG) and SBL(GiG) after preprocessing and postprocessing, respectively.

**Table 1 entropy-25-00511-t001:** Performance results of SBL(BGiG) for Cases 1–3.

Case 1: $\left(\alpha_{0}=0.01,\beta_{0}=0.99\right)$				Case 2: $\left(\alpha_{0}=0.1,\beta_{0}=0.9\right)$				Case 3: $\left(\alpha_{0}=1.4,\beta_{0}=2.0\right)$			
λ	$P_{D}$	$P_{\mathrm{FA}}$	**NMSE (dB)**	λ	$P_{D}$	$P_{\mathrm{FA}}$	**NMSE (dB)**	λ	$P_{D}$	$P_{\mathrm{FA}}$	**NMSE (dB)**
0.8	0.20	0	−2.367	0.8	0.24	0	−3.109	0.8	0.72	0	−16.264
0.6	0.08	0	−1.326	0.6	0.16	0	−2.197	0.6	1	0	−5.226
0.4	0.08	0	−1.181	0.4	0.08	0	−1.181	0.4	1	1	−0.088

**Table 2 entropy-25-00511-t002:** Settings for preprocessing analysis and simulations on SBL(BGiG).

$\alpha_{0}$	$\beta_{0}$	$a_{0}$	$b_{0}$	$\theta_{0}$	$\theta_{1}$	Sparsity	γ	*N*
[0.1,2]	[0.1,2]	$10^{-3}$	$10^{-3}$	$10^{-6}$	$10^{-6}$	25	(Equation 5)	100
s	τ	x	$x_{s}$	*M*	ϵ	e	*A*	y
(Equation 5)	(Equation 6)	(Equation 6)	$x_{s}=x\circ s$	5:N	316	(Equation 7)	${\left[A\right]}_{mn}∼N\left(0,1\right)$	$Ax_{s}+e$

**Table 3 entropy-25-00511-t003:** Settings for preprocessing analysis and simulations on SBL(GiG).

$a_{0}$	$b_{0}$	$\theta_{0}$	$\theta_{1}$	Sparsity	*N*
$10^{-3}$	$10^{-3}$	$10^{-6}$	$10^{-6}$	25	100
$\tau_{n}$	$x_{s}$	*M*	e	*A*	y
(Equation 22)	(Equation 25)	5:N	(Equation 23)	${\left[A\right]}_{mn}∼N\left(0,1\right)$	$Ax_{s}+e$

## Data Availability

https://github.com/MoShekaramiz/Compressive-Sensing-GiG-versus-BGiG-Modeling.

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
