# Peer review of "Compressive Sensing via Variational Bayesian Inference under Two Widely Used Priors: Modeling, Comparison and Discussion"

_entropy, 2023, doi:10.3390/e25030511_

Round 1

Reviewer 1 Report (New Reviewer)

The paper presents an interesting idea but should be modified in some of its parts:

- An overview figure of the proposed framework should be inserted, suitably commented and inserted in the text;

- A section relating to the state of the art is missing to set the focus on the problem faced;

- The algorithms on page 6 and 10 should be better contextualized in the text and referenced in terms of descriptions;

- A detailed description of the datasets used is missing. Maybe insert a table;

- What happens if the data is organized in a graph-structured way? A recent paper provides guidance on how to handle data in this case:

Manipur, Ichcha, Mario Manzo, Ilaria Granata, Maurizio Giordano, Lucia Maddalena, and Mario R. Guarracino. "Netpro2vec: a graph embedding framework for biomedical applications." IEEE/ACM Transactions on Computational Biology and Bioinformatics 19, no. 2 (2021): 729-740.

Author Response

Reviewer 2 Report (New Reviewer)

This manuscript discussed the problem of sparse signal recovery in compressive sensing via the variational Bayesian inference. Two priors, including Bernoulli-Gaussian-inverse Gamma and Gaussian-inverse Gamma, were particularly discussed. The performances of the Bayesian models using the aforementioned priors have been compared through simulations.

This manuscript seems to be an extended version of the authors’ previous publications, such as

M. Shekaramiz and T. K. Moon, “Compressive sensing via variational Bayesian inference,” In IEEE 2020 Intermountain Eng., Tech. and Comput. (IETC), pp. 1–6, 2020.

Major algorithms and theories in this manuscript can be found in the above earlier publication. The two priors discussed in this manuscript have been also compared in the above publication (see the end of Introduction of [65].

Considering that there were no new theory or algorithms found in the manuscript and the major finding, i.e., the performance difference of two priors, can be easily inferred from the earlier publication, I am afraid that I am not able to recommend this manuscript to be as a new journal publication.

Round 2

Reviewer 1 Report (New Reviewer)

As far as I'm concerned, no further changes are required

Reviewer 2 Report (New Reviewer)

The authors addressed my previous concerns. I have no further comments.

This manuscript is a resubmission of an earlier submission. The following is a list of the peer review reports and author responses from that submission.

Round 1

Reviewer 1 Report

The overall contribution of the paper is good. The novelty of the paper is sufficient. It needs a careful proofread. I have no hesitations to recommend this paper for publication.

Author Response

We want to thank the reviewer for their constructive comment on our manuscript. Accordingly, we have proofread the manuscript and modified it as there were some typos and grammatical mistakes in the original version. 

Reviewer 2 Report

The contribution of the current manuscript is not sufficient for publication in this Journal. Thus,  I regret to inform you that I have decided against publishing your manuscript

Author Response

We want to thank the reviewer for the time they put into reviewing our manuscript. 

Reviewer 3 Report

Because this is an empirical study paper, I think it's better to include a discussion about the "limitations" of this research. For example, the limited study from the synthetic dataset.

This paper studies the detailed behavior of recently proposed variational Bayesian inference algorithms from Bernoulli-Gaussian-inverse Gamma (BGig) and Gaussian-inverse Gamma (GiG). The main focus is comparing BGiG and GiG using a synthetic dataset and a discussion about the improvement of each method.

This paper is an empirical study rather than a theoretical development paper. The proposed research of many different scenarios is beneficial to understand the behavior of each algorithm. However, I am not fully convinced by the conclusion itself, meaning GiG is better than BGiG, mainly because the studied benchmark dataset from the synthetic data still needs to be improved. If the authors want to develop this methodological study further for the Bayesian community, I highly encourage them to release their studied code to the public. This way, the community could benefit better from this research paper and leverage the method in real scenarios for practitioners.

Minor Comments:

L162: "n" needs to be removed.

Figure 10 on page 13: the caption (c) $P_d-P_{FA}$ could be better placed.

Author Response

We want to appreciate the time the reviewer put into our work and their very constructive comments. We have modified the manuscript, accordingly. Below are the reviewer's comments and our responses and modifications.

1) "Comments and Suggestions for Authors: Because this is an empirical study paper, I think it's better to include a discussion about the "limitations" of this research. For example, the limited study from the synthetic dataset."

     Response: We thank the reviewer for their suggestion. We have modified the language in the conclusion section, as follows (Lines 273-277, page 14):

Based on our study on the synthetic data by considering the overall performance of both algorithms and the complexity in additional pre-/post-processing, we observed that SBL(GiG) is performing better than SBL(BGiG) modeling when using VB for sparse signals with no specific pattern in the supports. However, a rigorous comparison is required to study our work here under real-world scenarios. 

----------------------------------------

2) Comment: "This paper is an empirical study rather than a theoretical development paper. The proposed research of many different scenarios is beneficial to understand the behavior of each algorithm. However, I am not fully convinced by the conclusion itself, meaning GiG is better than BGiG, mainly because the studied benchmark dataset from the synthetic data still needs to be improved. If the authors want to develop this methodological study further for the Bayesian community, I highly encourage them to release their studied code to the public. This way, the community could benefit better from this research paper and leverage the method in real scenarios for practitioners."

    • Response: This is a great comment. We do agree that more rigorous analysis will be needed in various settings to draw a full conclusion. Our intention here was to look at the detail of each algorithm, show the problems observed in the VB inference for the two widely used models, look at some solutions to tackle the issues, and compare the models in the synthetic setting. Based on the comment, we have modified the conclusion section and also added the link through the GitHub repository, and released our codes to the public so that our methodological study can be used for further study by the community and practitioners. Please see L.276-278 in the revised manuscript, as shown below, as well.

"However, a rigorous comparison is required to study our work here under real-world scenarios. The MATLAB codes for GiG and BGiG modeling are available at
https://github.com/MoShekaramiz/Compressive-Sensing-GiG-versus-BGiG-Modeling.git."

----------------------------------------

3) "Minor Comments: 

L162: "n" needs to be removed.

Figure 10 on page 13: the caption (c) $P_d-P_{FA}$ could be better placed."

   Response: We thank the reviewer for their time and for detecting these issues.

We have proofread our manuscript, removed 'n' from L162, and modified the caption of Figure 10 on page 13. Please see L164 on page 9, and Figure 10 on page 13 of the revised manuscript.

Reviewer 4 Report

The paper describes a comparison of using variational inference for compressive sensing using two different priors, one which incorporates a sampling vector and the other which has a large noise prior. The comparison is done rigorously and results are shown well. The reviewer recommends this paper for publication after a few minor changes are made to figures. 

The legends and axis labels on some plots are too small to read. Please enlarge them for Figs 2 & 3

Several of the figure legends do not match the symbols in equations. For example, if the symbols is given as x_s in the text, the figure legend should be consistent. Please fix this where it occurs. 

The reviewer suggests a more detailed abstract, including a brief summary of conclusions, to help casual readers.